# *TET2* mutations are associated with hypermethylation at key regulatory enhancers in normal and malignant hematopoiesis

Morten Tulstrup [1,2,3], Mette Soerensen [4,5], Jakob Werner Hansen[1,2,3], Linn Gillberg [1,2], Maria Needhamsen[6], Katja Kaastrup[1,2,3], Kristian Helin[2,3,7], Kaare Christensen [4,5], Joachim Weischenfeldt [2,3,8 ✉] & Kirsten Grønbæk [1,2,3 ✉]

Mutations in the epigenetic modifier *TET2* are frequent in myeloid malignancies and clonal hematopoiesis of indeterminate potential (CHIP) and clonal cytopenia of undetermined significance (CCUS). Here, we investigate associations between *TET2* mutations and DNA methylation in whole blood in 305 elderly twins, 15 patients with CCUS and 18 healthy controls. We find that *TET2* mutations are associated with DNA hypermethylation at enhancer sites in whole blood in CHIP and in both granulocytes and mononuclear cells in CCUS. These hypermethylated sites are associated with leukocyte function and immune response and ETS-related and C/EBP-related transcription factor motifs. While the majority of *TET2*-associated hypermethylation sites are shared between CHIP and in AML, we find a set of AML-specific hypermethylated loci at active enhancer elements in hematopoietic stem cells. In summary, we show that *TET2* mutations is associated with hypermethylated enhancers involved in myeloid differentiation in both CHIP, CCUS and AML patients.

[1] Department of Hematology, Rigshospitalet, Copenhagen, Denmark. [2] Biotech Research & Innovation Centre (BRIC), University of Copenhagen, Copenhagen, Denmark. [3] Novo Nordisk Foundation Center for Stem Cell Biology, DanStem, Faculty of Health Sciences, University of Copenhagen, Copenhagen, Denmark. [4] The Danish Twin Registry, University of Southern Denmark, Odense, Denmark. [5] Epidemiology, Biostatistics and Biodemography, Department of Public Health, University of Southern Denmark, Odense, Denmark. [6] Center for Molecular Medicine, Department of Clinical Neuroscience, Karolinska Institutet, Stockholm, Sweden. [7] Cell Biology Program and Center for Epigenetics Research, Memorial Sloan Kettering Cancer Center, New York, NY, USA. [8] Finsen Laboratory, Rigshospitalet, Copenhagen, Denmark. ✉email: joachim.weischenfeldt@bric.ku.dk; kirsten.groenbaek@regionh.dk

Mutations in genes encoding regulators of DNA methylation are frequent in hematological cancers such as myelodysplastic syndromes (MDS), acute myeloid leukemia (AML), and lymphoma[1–4]. Two recurrently mutated genes with opposing effects on DNA methylation are ten-eleven translocation methylcytosine dioxygenase 2 (TET2) and DNA methyltransferase 3A (DNMT3A), both of which are especially frequently mutated in myeloid neoplasms and in the related conditions clonal hematopoiesis of indeterminate potential (CHIP) and clonal cytopenia of undetermined significance (CCUS). CHIP is characterized by clonal propagation of blood cells carrying driver mutations in known myeloid cancer related genes in otherwise hematologically healthy individuals and is associated with an increased risk of hematological cancer and cardiovascular disease[5,6]. The latter is suspected to be caused by a hyperinflammatory phenotype of cells (especially monocytes and macrophages) with CHIP mutations, which have an increased production of proinflammatory cytokines such as interleukin (IL)-1β and IL-6[7,8]. Furthermore, CHIP mutations by definition confer an increased capacity for self-renewal, and TET2 mutations have been shown to skew hematopoietic progenitor cells toward myelomonocytic differentiation[9]. CCUS is characterized by the same mutations as CHIP along with cytopenia and absence of dysplastic bone marrow features and is associated with a high risk of developing overt myeloid malignancy. The mechanisms by which CHIP and CCUS mutations lead to phenotypic alterations at the cellular and clinical level are poorly understood. TET2 and DNMT3A have different functions in regulating DNA methylation, with TET2 catalyzing the conversion of 5-methylcytosine to 5-hydroxymethylcytosine, the initial step in DNA demethylation[10], while DNMT3A catalyzes de novo methylation of cytosine residues[11]. A likely explanation for the altered phenotype is that loss-of-function mutations in TET2 and DNMT3A confer specific changes to the DNA methylome, which in turn lead to altered regulation of specific genes or gene sets. However, while the specific DNA methylation alterations conferred by TET2 and DNMT3A mutations have been described in human cancer and in cell lines and mouse models with gene inactivation[4,9,12–15], somatic mutations in these genes have not been investigated in vivo in hematologically healthy human individuals.

Here, we show that TET2 mutations are associated with a distinct pattern of whole blood DNA hypermethylation in a cohort of elderly twins sampled from the general population. This hypermethylation pattern is also detectable in both granulocytes and mononuclear cells from patients with TET2-mutated CCUS. The same pattern is present in AML, but occurs together with additional changes, indicating that TET2 may be involved in regulating additional methylation sites in AML compared to CHIP and CCUS.

## Results

We analyzed DNA methylation data from 305 individuals in the Danish Twin Registry, of whom 116 had one or more CHIP mutations, including 55 with a DNMT3A mutation and 44 with a TET2 mutation (Supplementary Figs. 1–3 and Supplementary Table 1). Methylation profiles were characterized using the Illumina 450k array, and 427,112 CpG sites passed quality control. None of the individuals with TET2 or DNMT3A mutations were diagnosed with a myeloid disease in the national Danish Patient Registry. There were no differences in global average beta methylation levels between individuals with vs without CHIP, or in individuals with TET2 or DNMT3A mutations compared with individuals without CHIP (Supplementary Fig. 4). We analyzed differences in blood cell-type proportions inferred from DNA methylation levels and found that TET2 mutations were associated with higher proportions of monocytes ($P = 0.012$), but otherwise there were no significant differences in cell proportions between individuals with vs. without TET2 or DNMT3A mutations (Supplementary Fig. 5). The observed higher monocyte counts associated with TET2 mutations are consistent with experimental data from mice[16] and observational data from humans[17].

**TET2 mutations are associated with widespread CpG hypermethylation in CHIP.** We next investigated the association between DNMT3A mutations and methylation levels in an epigenome-wide fashion by comparing the 55 individuals with DNMT3A mutations to the 189 individuals without any CHIP mutations while adjusting for age, sex, batch effects, four principal components, and blood cell proportions inferred from methylation profiles, and observed moderate inflation of the P values (genomic inflation factor, $\lambda = 1.12$, Fig. 1a and Supplementary Data 1). When performing a similar analysis of the 44 individuals with TET2 mutations, we found the genomic inflation to be considerably larger, indicating that TET2 mutations may affect methylation levels at a large number of loci ($\lambda = 1.29$, Fig. 1b and Supplementary Data 2). A volcano plot of the results from the TET2-specific analysis revealed that the CpG sites with low P values were almost exclusively positively associated with TET2 mutation status (Fig. 1c). We defined hypermethylated sites as sites with a positive association with TET2 mutations at the significance threshold where 95% of CpG sites were hypermethylated (Fig. 1d). At $P < 1.4 \times 10^{-5}$, 2741 out of 2885 sites were hypermethylated, and we selected these 2741 sites for further investigation. We tested whether the methylation level at the 2741 sites was also positively correlated with TET2 VAF and found this to be the case for 2694 (98.3%) of these sites ($P < 1 \times 10^{-16}$, Supplementary Fig. 6). Since both CHIP and DNA methylation have been shown to correlate with smoking, we carried out sensitivity analyses with adjustment for cumulative tobacco exposure and current smoking and observed no meaningful changes to the results (all of the 2741 hypermethylated sites remained positively associated with TET2 mutations, 95% of them remained below the significance threshold in both analyses, and the highest observed P value after correction was $4.3 \times 10^{-5}$, Supplementary Methods 1 and Supplementary Fig. 7). A Manhattan plot showed that the 2741 sites are not restricted to one or a few specific loci but distributed across the genome (Fig. 1e). The DNMT3A-specific analyses identified 160 CpG sites that were hypomethylated in DNMT3A mutation carriers (Supplementary Figs. 8 and 9). Again, we found no evidence of confounding from smoking (Supplementary Fig. 10). In summary, these results demonstrate that the presence of CHIP mutations in TET2 are associated with increased CpG methylation levels at a large number of genomic loci, while DNMT3A mutations only led to detectable changes at relatively few loci. This hypermethylation phenotype is most likely directly associated with TET2 mutations and not due to changes in cell type composition, since this was adjusted for in the model.

**Gene ontology enrichment analysis and chromatin states at hypermethylated sites.** Gene Ontology enrichment analysis revealed that the 2741 hypermethylated sites are located in or near genes involved in a number of biological processes all related to immune response and (especially myeloid) leukocyte function (Fig. 2a). No enriched Gene Ontology terms were identified using the results from the DNMT3A-specific analyses. We next compared our findings with chromatin states in monocytes using ChromHMM data from Roadmap Epigenomics[18] and found that close to a third (31%) of the 2741 hypermethylated sites were

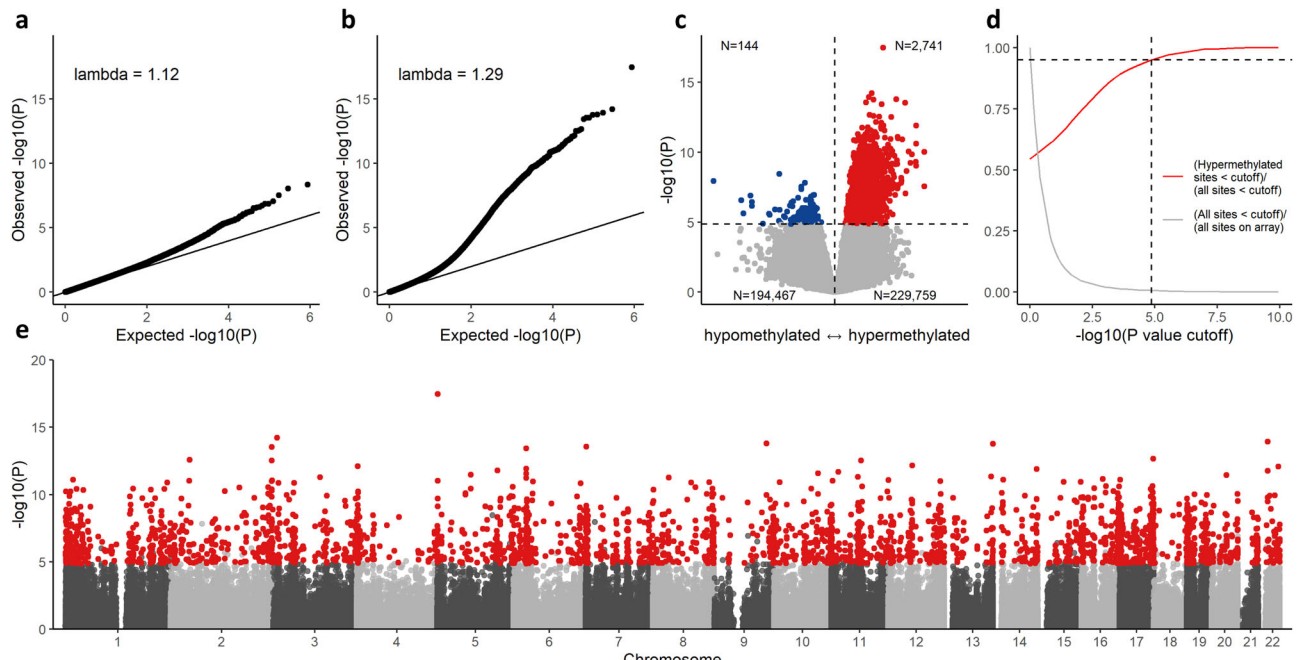

**Fig. 1 Epigenome-wide association studies of *DNMT3A* and *TET2* mutations. a** QQ plot the *P* value distribution in differential methylation analyses by *DNMT3A* mutation status (55 *DNMT3A*-mutated inviduals compared to 189 individuals without CHIP). **b** Same as A, but comparing 44 *TET2*-mutated individuals to 189 without CHIP. **c** Volcano plot of results from *TET2*-specific analyses. Horizontal dashed line indicates $P = 1.4 \times 10^{-5}$. Numbers in corners indicate the number of CpG sites in each quadrant separated by dashed lines. Red and blue dots represent hyper- and hypomethylated CpG sites, respectively, below the significance threshold. **d** Selection of *P* value cutoff used in C. Red line displays the increasing proportion of hypermethylated sites with decreasing *P* value cutoff. Horizontal dashed line indicates 95% of sites are hypermethylated. Vertical dashed line indicates selected *P* cutoff of $1.4 \times 10^{-5}$. **e** Manhattan plot with the 2741 CpG sites showing *TET2* mutation-associated hypermethylation highlighted in red. *P* values and effect size estimates in all panels derived using a linear mixed effects regression with twin pair as random intercept. All *P* values are two-sided and not adjusted for multiple comparisons.

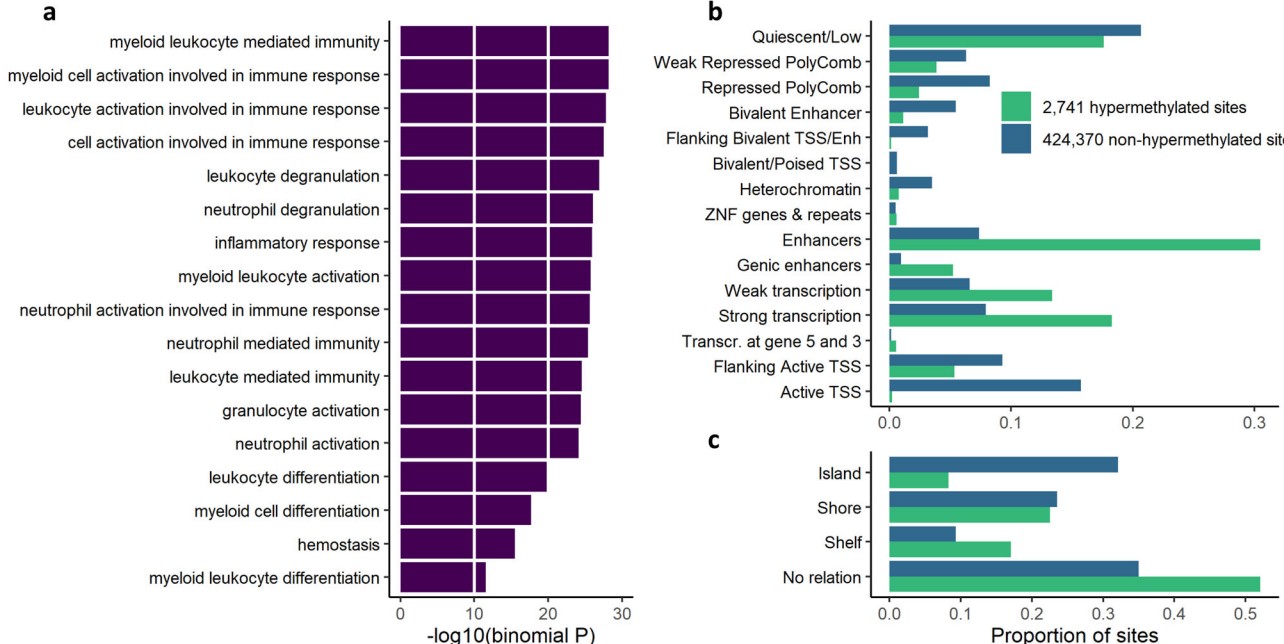

**Fig. 2 Characteristics of the 2741 CpG sites hypermethylated in *TET2*-mutated CHIP. a** Enrichment of Gene Ontology terms for 2741 hypermethylated sites. *P* values derived using the region-based binomial test in GREAT and are not adjusted for multiple comparisons. **b** Monocyte chromatin states at hypermethylated and non-hypermethylated sites. **c** CpG island relations for hypermethylated and non-hypermethylated sites.

located in active enhancer regions, while this was the case for only 7% of the non-hypermethylated sites ($P < 10^{-16}$, $\chi^2$ test, Fig. 2b). Similarly, regions annotated as genic enhancers and regions of active transcription were highly enriched among the 2741 CpG sites, while other regions such as repressed polycomb and quiescent chromatin were underrepresented. Furthermore, while 32% of the non-hypermethylated CpG sites were located in CpG islands, this was the case for only 8% of the 2741 hypermethylated sites ($P < 10^{-16}$, $\chi^2$ test), which conversely were overrepresented in genomic regions with low CpG density (Fig. 2c). These observations are consistent with previous reports that TET2 binds specifically at enhancer sites in e.g. myeloid mouse cells[19]. Furthermore, TET2 mutations do not appear to affect methylation status at CpG islands in cancer[12], likely due to strict control of the methylation of CpG islands by mechanisms involving TET1 and H3K4me3[20].

In conclusion, we find that TET2 associated CpG hypermethylation can already be identified in the premalignant state in individuals with CHIP, and that these CpG sites are occurring predominantly at enhancer regions.

**Replication of *TET2* mutation-associated DNA methylation signature in patients with CCUS.** To investigate whether *TET2* mutations have similar effects on DNA methylation in CCUS, we carried out DNA methylation profiling in DNA from peripheral blood granulocytes from five patients with CCUS with large *TET2*-mutated clones (VAF: 33%, 45%, 47%, 50%, and 50%) and in eight healthy controls (Supplementary Table 2), this time using the larger Illumina 850k EPIC array, which has a better coverage of enhancer regions and non-CpG island sites. Using our definition of *TET2* mutation-associated CpG hypermethylated sites, we found 12,096 CpG sites to be significantly hypermethylated (Fig. 3a and Supplementary Data 3). Of the 2741 CpG sites which were hypermethylated in CHIP, 2460 were available for evaluation in the CCUS data after quality control, and of these 2427 (99%) were positively associated with *TET2* mutations in CCUS ($P < 10^{-16}$, $\chi^2$ test). Similarly, 93% of the significantly hypermethylated sites in CCUS were positively associated with *TET2* mutations in CHIP ($P < 10^{-16}$, $\chi^2$ test, Fig. 3b). Gene ontology enrichment analysis of the 12,096 hypermethylated sites identified many of the same biological processes related to myeloid leukocyte function as the CHIP-based analysis (Fig. 3c). Moreover, the proportions of enhancers and regions of low CpG density were strikingly similar to the results from the analyses on CHIP (Fig. 3d, e).

To investigate whether the observed hypermethylation was present in other lineages than granulocytes, we analyzed DNA from T-cell depleted, bone marrow-derived mononuclear cells (MNCs) from 20 patients with CCUS, of whom 10 had a *TET2* mutation (median VAF: 40% [interquartile range 38–41%], Supplementary Table 3). Among sites that were significantly hypermethylated in CHIP, 99.6% were positively associated with *TET2* mutations in CCUS MNCs ($P < 10^{-16}$, $\chi^2$ test, Supplementary Fig. 11A and Supplementary data 4). Correspondingly, 96.3% of significantly hypermethylated sites in CCUS granulocytes were positively associated with *TET2* mutations in CCUS MNCs ($P < 10^{-16}$, $\chi^2$ test, Supplementary Fig. 11B). Gene ontology enrichment analysis, and analysis of chromatin states and CpG island relations revealed similar results as analyses of granulocyte DNA (Supplementary Fig. 11C–E).

Methylation at transcription start sites can play an important regulatory role in cancer[21]. To this end, we identified regions of hypermethylation at TSS sites in in CHIP and CCUS. Only one hypermethylated TSS sites overlapped between both CHIP and the two CCUS datasets (Supplementary Methods 2), and this site

does not have a known or suspected role in cancer (Supplementary Table 4).

In summary, we find a high concordance between *TET2* mutation-associated CpG hypermethylation in CHIP and CCUS. The hypermethylation pattern is clearly detectable in both whole blood, peripheral blood granulocytes, and bone marrow-derived MNCs. Together with the observation that the mutated clones in the CCUS granulocyte samples constituted at least 90% of blood cells in four out of five patients, these results further support our conclusion that *TET2* mutations are causally linked with the observed hypermethylation in the premalignant state.

**TET2-associated hypermethylation co-localizes with specific transcription factor motifs.** Since TET2 activity is known to occur in the vicinity of different transcription factor binding sites in different cellular contexts, we next carried out motif enrichment analyses of enhancer regions with *TET2* mutation-associated hypermethylation in CHIP and CCUS. We defined 200 bp regions centered around each of the CpG sites located in monocyte enhancers and ranked them according to the degree of hypermethylation in CHIP and CCUS (granulocyte data), respectively. Of 678 unique transcription factors analyzed, 46 were significantly enriched in CHIP and 75 were enriched in CCUS (Supplementary Data 5). Of these, 28 were enriched in both CHIP and CCUS ($P < 1 \times 10^{-16}$, $\chi^2$ test). Most notably, a large number of members of the ETS transcription factor family, including ELF2, ETS1, and the master hematopoietic regulator SPI1, were strongly enriched in both CHIP and CCUS (Fig. 4). Of 25 ETS transcription factor motifs, 21 (84%) were enriched in CHIP ($P < 1 \times 10^{-16}$, Fisher's exact test) and 23 (92%) were enriched in CCUS (Fisher's exact test $P < 1 \times 10^{-16}$). Furthermore, seven out of ten (70%) C/EBP-related transcription factors were enriched in CHIP (Fisher's exact test $P = 0.03$) and nine (90%) out of ten were enriched in CCUS (Fisher's exact test $P = 1 \times 10^{-8}$).

In summary, we find a strong enrichment of ETS and C/EBP transcription factor binding motifs at or near *TET2* mutation-associated CpG hypermethylation in both CHIP and CCUS, suggesting that TET2 activity is directly linked with recruitment of key transcription factors instructive for cellular identity and differentiation in hematopoiesis.

**TET2 mutations in AML are associated with increased methylation levels at hematopoietic stem cell-specific enhancers.** To investigate whether *TET2* mutation-associated hypermethylation is consistent between AML and CHIP and CCUS, we analyzed data from 90 AML patients with available sequencing and 450k methylation array profiling from The Cancer Genome Atlas (TCGA) LAML cohort, of whom four had a *TET2* mutation. At $P < 0.01$, 2000 sites were hypermethylated, constituting 72% of all sites below this significance cutoff ($P = 8.2 \times 10^{-12}$, $\chi^2$ test).

Of the 2741 sites which were hypermethylated in CHIP, 2310 were available in the TCGA dataset, and 1895 (82%) of these were positively associated with *TET2* mutations ($P < 1 \times 10^{-16}$, $\chi^2$ test, Fig. 5a). Similarly, 78% of the sites that were hypermethylated in CCUS granulocytes were also hypermethylated in AML ($P < 1 \times 10^{-16}$, $\chi^2$ test, Fig. 5b). Only 54% of the 2000 hypermethylated sites in AML were positively associated with *TET2* mutations in CHIP ($P = 0.81$, $\chi^2$ test, Supplementary Fig. 12). In contrast to this, 72% the 2000 hypermethylated sites in AML were also positively associated with *TET2* mutations in CCUS granulocytes ($P < 1 \times 10^{-16}$, $\chi^2$ test, Fig. 5c). Together, these results show that while the hypermethylation observed in CHIP is also detectable in AML, many additional sites are also hypermethylated in AML, and these sites are not

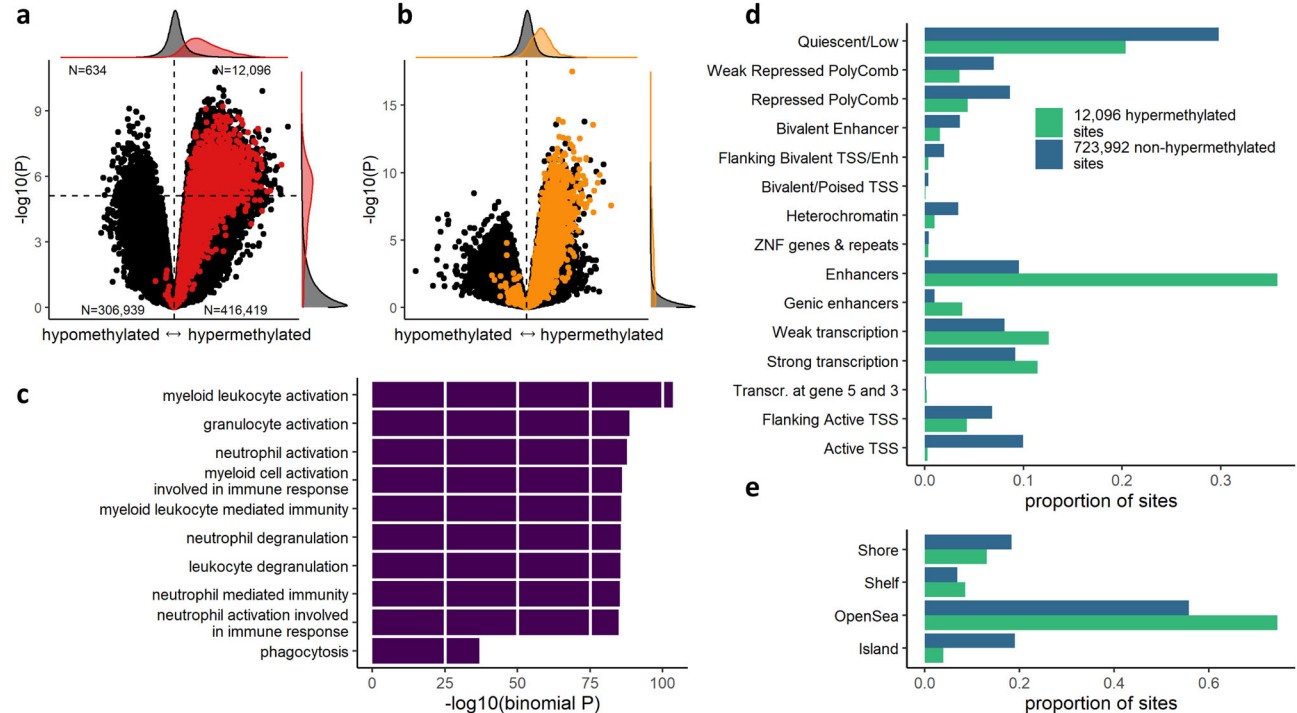

**Fig. 3 Results of epigenome-wide analyses of *TET2* mutations in granulocyte DNA from five CCUS patients and eight healthy controls. a** Volcano plot of CCUS results. Red dots indicate CpG sites from the set of 2741 hypermethylated sites in CHIP (Fig. 1c). Margins display distributions of the 2741 sites versus the rest of the EPIC array. Numbers in corners indicate the number of dots (both colors) in each quadrant separated by dashed lines. *P* values and effect size estimates derived using a linear model in Limma. *P* values are two-sided and not adjusted for multiple comparisons. **b** Volcano plot of CHIP results (same as 1C), yellow dots indicate CpG sites from the set of 12,096 significantly hypermethylated sites in CCUS granulocytes (upper right quadrant in panel **a**). Only sites analyzed in both CHIP and CCUS are shown. *P* values and effect size estimates derived using a linear model in Limma. *P* values are two-sided and not adjusted for multiple comparisons. **c** Enrichment of Gene Ontology terms for 12,096 hypermethylated sites in CCUS. *P* values derived using the region-based binomial test in GREAT and are not adjusted for multiple comparisong. **d** Monocyte chromatin states for hypermethylated and non-hypermethylated sites. **e** CpG island relations for hypermethylated and non-hypermethylated sites.

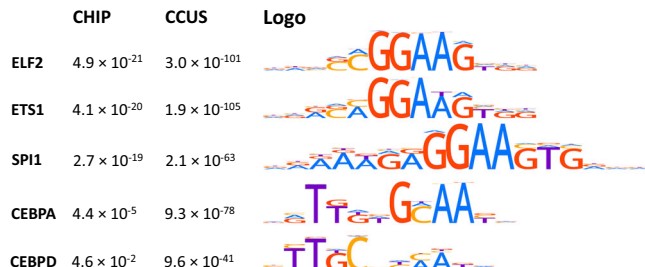

**Fig. 4 Selected enriched transcription factor binding motifs at hypermethylated monocyte enhancer regions in CHIP and CCUS.** Hypermethylated regions defined as 200 bp flanking regions around CpG sites located in an enhancer region. Enrichment (E) values derived using the Fischer's exact test method in Analysis of Motif Enrichment. E values are adjusted *P* values with correction for the number of tested transcription factor binding motifs from the HOCOMOCO v11 full dataset (*N* = 769).

hypermethylated in CHIP. The sites that are hypermethylated in AML are more likely to also be hypermethylated in CCUS, indicating that although the pattern of *TET2* associated hypermethylation in CHIP and CCUS is highly similar, CCUS also shows additional hypermethylation features that are more similar to those observed in AML.

Gene Ontology enrichment analysis of the 2000 hypermethylated sites did not identify any significantly enriched biological processes. Furthermore, CpG sites located in monocyte enhancers were only slightly enriched among the 2000 hypermethylated sites

(9% vs 7% of non-hypermethylated sites, $P = 7 \times 10^{-5}$, $\chi^2$ test, Fig. 5d). Since chromatin states in monocytes are likely not a good proxy for AML blasts, we also calculated the proportion of hypermethylated sites in AML that were located in enhancers from the Roadmap Epigenomics ChromHMM annotation of $CD34^+$ hematopoietic stem cells (HSCs), and found a higher degree of enrichment for enhancer regions (15% in HSCs vs 8% in monocytes, $P < 1 \times 10^{-16}$, $\chi^2$ test). We then compared average *TET2* mutation-associated methylation differences for both CHIP, CCUS, and AML between sites in regions annotated as monocyte enhancers but not HSC enhancers (monocyte-specific enhancers) and sites in regions annotated as HSC enhancers but not monocyte enhancers (HSC-specific enhancers), and found that while only monocyte-specific enhancers were hypermethylated in CHIP and in CCUS (both $P < 1 \times 10^{-16}$, Student's *t* test, Fig. 5e and Supplementary Fig. 13), HSC-specific enhancers were significantly more hypermethylated than monocyte-specific enhancers in AML ($P = 5.7 \times 10^{-10}$, Student's *t* test). Motif enrichment analysis of *TET2* hypermethylated HSC enhancer regions revealed overrepresentation of 238 transcription factors. As in CHIP and CCUS, ETS transcription factors were overrepresented (84% of motifs significantly enriched, $P = 5.6 \times 10^{-7}$, $\chi^2$ test), but not C/EBP factors ($P = 0.18$). In conclusion, these results suggest that while *TET2* mutations in AML are associated with many of the same methylation changes as in CHIP and CCUS, they are also associated with hypermethylation at a large number of other sites, especially at sites associated with active enhancers in HSCs but not in mature monocytes. Similarly, for *DNMT3A* mutations, the few

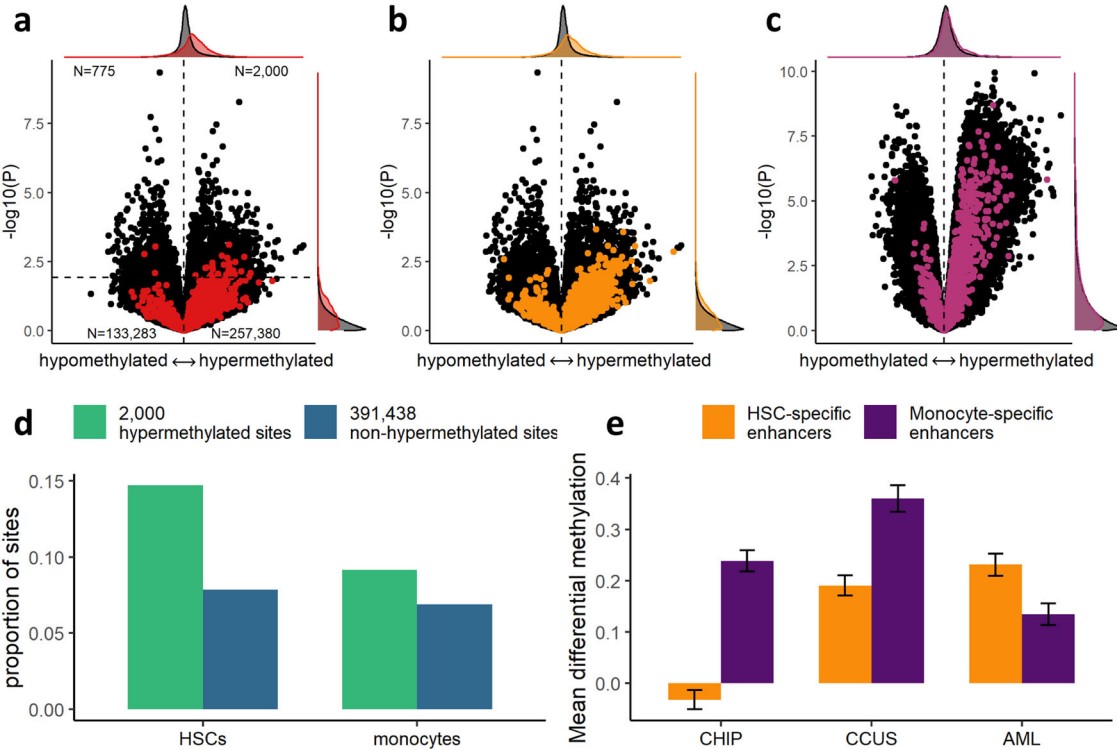

**Fig. 5 *TET2* mutation-associated hypermethylation in TCGA LAML data. a** Results of epigenome-wide association studies in four and 86 AML patients with and without *TET2* mutations, respectively. Red dots indicate CpG sites from the set of 2741 significantly hypermethylated sites in CHIP (Fig. 1c). Horizontal dashed line indicates $P = 0.01$. Margins display distributions of the 2741 sites versus the rest of the sites. Numbers in corners indicate the number of dots (both colors) in each quadrant separated by dashed lines. *P* values and effect size estimates derived using a linear model in Limma. *P* values are two-sided and not adjusted for multiple comparisons. **b** Similar to **a**. Orange dots indicate CpG sites from the set of 12,096 significantly hypermethylated sites in CCUS granulocytes. Only overlapping sites between the two sets are shown. *P* values and effect size estimates derived using a linear model in Limma. *P* values are two-sided and not adjusted for multiple comparisons. **c** Volcano plot of CCUS granulocyte results (same as in Fig. 3a) but with the 2000 significantly hypermethylated sites in AML highlighted (purple dots). *P* values and effect size estimates derived using a linear model in Limma. *P* values are two-sided and not adjusted for multiple comparisons. **d** Proportion of hypermethylated and non-hypermethylated sites in AML located in hematopoietic stem cell (HSC)-specific ($N = 14,398$) and monocyte-specific enhancers ($N = 14,017$). **e** Mean scaled effect size estimates of *TET2* mutations at all CpG sites located in HSC-specific ($N = 47,748$) and monocyte-specific enhancers ($N = 37,518$). Error bars indicate 95% confidence intervals. See Supplementary Fig. 13 for plots of data distributions.

methylation changes observed in CHIP were likely to also be present in AML, but at the same time many more changes were present in AML that were not detected in CHIP.

**DNMT3A mutations are associated with widespread hypomethylation in AML in regions overlapping with *TET2* hypermethylation.** We observed significant correlation between *DNMT3A* mutation and CpG hypomethylation in the TCGA AML cohort, with 19,235 hypomethylated sites reaching statistical significance at $P < 0.018$ (Supplementary Fig. 14). The 160 CpG sites that were hypomethylated in *DNMT3A*-mutated CHIP correlated with *DNMT3A* mutation status in AML (125 [94%] of 133 CpG sites present in both datasets, $P = 4.1 \times 10^{-13}$, $\chi^2$ test). The 19,235 hypomethylated sites in AML were also enriched in enhancer regions (20% compared to 7% of the remaining sites, $P < 1 \times 10^{-16}$, $\chi^2$ test, Supplementary Fig. 15). We asked whether and to what extent mutations in *DNMT3A* and *TET2* impacted the methylation status at the same CpG sites. In CHIP, CpG sites that were negatively associated with *DNMT3A* mutations were underrepresented among the 2741 *TET2*-associated hypermethylated CpG sites (48% compared to 54% of other sites, $P = 6 \times 10^{-10}$, $\chi^2$ test Supplementary Fig. 16), but in AML the 2000 *TET2*-associated hypermethylated sites were enriched for hypomethylation in DNMT3A-mutated tumors ($P < 1 \times 10^{-16}$, $\chi^2$ test). There were no AML patients with both *TET2* and *DNMT3A* mutations, suggesting

mutual exclusivity. In conclusion, *DNMT3A* mutations in AML are associated with a strong hypomethylation profile, including the few sites hypomethylated in *DNMT3A*-mutated CHIP. AML, but not CHIP, showed association between CpG sites that were hypermethylated in *TET2* mutated samples but hypomethylated in *DNMT3A*-mutated samples.

**Prediction of *TET2* mutations in CHIP and CCUS.** Our findings demonstrate a strong link between *TET2* mutation status and CpG hypermethylation at sites associated with gene-regulatory function already at the pre-malignant state. We next asked to what extent methylation status was predictive of *TET2* mutation status. To this end, we developed a DNA methylation-based prediction model of *TET2* mutation status. Such a model could in principle be useful in identifying carriers with *TET2* mutations using only DNA methylation information. We first used leave-one-out cross validation of an elastic net logistic regression model to predict the 44 *TET2* mutations in the CHIP cohort. While this method accurately predicted only 11 out of 44 *TET2* mutated cases corresponding to a sensitivity of 25%, all 261 individuals without *TET2* mutations were classified correctly, corresponding to a specificity of 100% (Supplementary Table 5 and Supplementary Fig. 17). In *TET2* mutation carriers, the predicted probability was positively correlated with clone size ($P = 2.3 \times 10^{-6}$). We next used the same model based on all

individuals in the CHIP cohort to predict *TET2* mutations in the two CCUS cohorts, and accurately classified all eight healthy controls and ten CCUS patients without *TET2* mutations (specificity 100%) and nine out of 15 cases, corresponding to a sensitivity of 60% (Supplementary Table 6 and Supplementary Fig. 18).

In summary, our results indicate that a methylation-based model can be used to predict *TET2* mutation status even with a modest cohort size.

## Discussion

The changes in DNA methylation that result from somatic *TET2* mutations have been studied in cancer and in mouse models, but whether and to what extent these mutations cause DNA methylation changes in CHIP and CCUS has not been addressed. The present study shows that *TET2* mutations in individuals with CHIP and CCUS are associated with DNA hypermethylation in a manner that is at the same time global and non-random, i.e. specific to enhancer elements, non-CpG island regions, and genes involved in myeloid leukocyte function and immune response. This is in line with previous studies in other organisms and diseases. A study of a *TET2*-mutated AML mouse model found that loss of *TET2* function in preleukemic cells resulted in enhancer-specific hypermethylation[13]. Other studies have found similar results in mouse embryonic stem cells[22] and human chronic myelomonocytic leukemia cells[12]. Another study, which applied chromatin immunoprecipitation with massively parallel sequencing (ChIP-seq) in myeloid mouse cells immortalized by AML1-ETO to investigate TET2 binding regions and subsequently investigated for enriched gene ontology terms, found several terms which were identical or similar to the ones found in our study, including "leukocyte activation" and "immune system process"[19]. This study also found many of the same motifs to be enriched at TET2 binding sites as we found for enhancers that were hypermethylated in *TET2* mutation carriers. Additionally, rare cases of germline *TET2* mutations have been linked to increased enhancer methylation especially at RUNX1 and SPI1 binding sites[23]. In conclusion, the present findings are in line with previous studies of *TET2* mutations in model organisms and human cancer, as well as human germline *TET2* haploinsufficiency. However, as TET2 function is known to be highly context-dependent, our study provides further insights both into the role of native TET2 in normal human hematopoiesis and the impact of *TET2* mutations in CHIP, CCUS, and AML. The DNA methylation signature of *TET2* mutations in non-malignant human cells is of special interest because they provide insight into how TET2 contributes to maintaining normal hematopoiesis through demethylation of specific regulatory elements[10]. The exact mechanism with which TET2 is recruited to genomic regions remains unclear since TET2 does not itself have any known DNA-binding domain. Rather, the protein is likely recruited to DNA through interaction with a range of different DNA-binding proteins such as transcription factors, with different factors involved in the process dependent on the specific cellular context. Interestingly, we found that of the transcription factors whose motifs were overrepresented among hypermethylated enhancer elements many were also identified in the aforementioned ChIP-seq study on myeloid mouse cells[19], including the ETS-related family of transcription factors which, among others, contains SPI1, a key regulator of hematopoietic differentiation which is especially important in monocyte and macrophage development[24]. Conversely, the AGATAA-containing motifs for transcription factors such as GATA1, GATA3, and EVI1 which were highly enriched in *TET2* binding sites in mouse preleukemic cells were not found to be enriched in our data,

indicating divergent roles in TET2 recruitment for these factors in the two species and/or cellular contexts.

An alternative hypothesis to explain the *TET2* mutation-driven alterations in cellular phenotype and differentiation was recently presented by Izzo et al.[9] who proposed that the skewed cellular differentiation conferred by *TET2* and *DNMT3A* mutations are a direct result of the CpG density of the transcription factor motifs and their consequent sensitivity to random DNA methylation or demethylation conferred by loss-of-function of each gene. One of the assumptions in this hypothesis is that the DNA methylation changes conferred by *DNMT3A* and *TET2* mutations are global and stochastic, which appears contradictory to the findings in the current study that *TET2* mutation-dependent hypermethylation is non-random and co-localizes with CpG low density regions, such as the CpG-poor binding motif of SPI1. Interestingly, Izzo et al. specifically showed that SPI1 motif accessibility as determined by single-cell ATAC-seq was not affected by *TET2* or *DNMT3A* mutations. Thus, while it is likely that some of the effects of *TET2* mutations on the cellular phenotype are the result of differences in transcription factor motif CpG content, another mechanism is the *TET2* mutation-dependent hypermethylation at especially enhancer regions, which results from selective and context-dependent recruitment of TET2 by key transcriptional regulators.

Using TCGA data we show that *TET2* mutations in AML are associated with hypermethylation at many of the same sites as in CHIP and CCUS, but also at a large number of AML-specific sites, which are unaffected in CHIP and CCUS, especially in regions that are enhancers in HSCs but not in monocytes. This suggests that the hypermethylation observed in CHIP may represent an early initiating event in the development towards malignant transformation. The additional hypermethylation at new sites in AML possibly reflects stem-like transcriptional regulation due to the presence of modifiers that are specific to leukemic blasts.

Our use of whole blood DNA methylation in the CHIP analyses has its limitations. Although the CHIP results are adjusted for estimated cell-type compositions, we cannot exclude the possibility of unmeasured confounding from a shift in cell proportions not detected by the deconvolution algorithm. However, the results observed in CHIP were replicated in both CCUS granulocytes and MNCs, strongly suggesting that these methylation patterns are independent of cellular lineage. The surprisingly low number of observed hypomethylated CpGs in *DNMT3A*-mutated CHIP may also be due to our use of unsorted, peripheral blood. CHIP mutations are likely to have lineage-specific consequences, and future investigations of CHIP-associated methylation changes in both stem cells and different hematopoietic lineages are needed to better understand the epigenetic consequences of CHIP mutations in humans.

We attempted to construct a prediction model of *TET2* mutations since such a model could have widespread applications in epigenetic research. Firstly, it would allow for instant imputation of *TET2* mutations in the many existing DNA methylation datasets from various cohorts and enable epidemiological investigations of associations between *TET2* mutations and a range of phenotypes without the need for targeted sequencing. Secondly, it would allow researchers to adjust for *TET2* mutations in epigenome-wide association studies, where they are a potential confounder of associations between DNA methylation and other phenotypes such as cardiovascular disease and age. While we were not able to robustly predict *TET2* mutation status in CHIP from DNA methylation levels alone, potentially due to the small cohort size, DNA methylation levels in non-mutated individuals were highly predictive of absence of *TET2* mutations, emphasizing the substantial differences in DNA methylation levels in

non-mutated vs *TET2*-mutated blood cells in these elderly individuals sampled from the general population.

Finally, since the *TET2* mutation-associated hypermethylation signature observed in our study was found to be associated with *TET2* VAF in CHIP, we suggest that methylation levels at these sites can be used to investigate the effect of epigenetic therapies on TET2 function. Studies in mice have shown that ascorbate deficiency can inhibit TET2 function, and it has been suggested that ascorbate supplementation in individuals with *TET2*-mutated CHIP or hematological diseases could help maximize the function of the remaining functional *TET2* allele[25,26], but this has not been demonstrated in humans. We suggest that methylation levels at the *TET2* mutation-associated hypermethylated sites can be used in future studies to investigate the effect of ascorbate and other epigenetic therapies such as hypomethylating agents on TET2 function.

## Methods

**Study subjects**. CHIP cases and controls were from the Longitudinal Study of Aging Danish Twins (LSADT)[27,28] which consists of twins aged 70 or more recruited between 1995 and 1997, and all blood samples were collected in 1997. All participants provided written informed consent prior to participation, and the study was approved by The Regional Committees on Health Research Ethics for Southern Denmark (S-VF-20040241 and S-20170053). Information on known myeloid disease was collected by extracting ICD-10 codes from the national Danish Patient Registry, which contains all hospital discharges and outpatient visits from all Danish Hospitals since 1977 and has been coded according to ICD-10 since 1994.

CCUS patients and healthy controls were recruited from the treating hematological departments in Denmark. The study was approved by the Danish National Committee on Research Ethics. All participants provided written informed consent prior to participation, and the study was conducted in accordance with Danish ethical regulation for work with human participants.

**DNA sequencing**. CHIP and CCUS mutations were detected using a targeted sequencing panel (Illumina TruSeq Custom Amplicon, Illumina, San Diego, CA, USA) covering 21 of the most commonly mutated genes in CHIP (Supplementary Table 7 and Supplementary Data 6). Libraries were prepared using 100–200 ng DNA. We used unique molecular identifiers consisting of six random nucleotides to optimize variant calling, and libraries were sequenced on the Illumina NextSeq platform using a 300-cycle output kit. Reads were aligned to hg19 using BWA-MEM v. 0.7.10[29], and variant calling was performed using FreeBayes v.1.1.0[30] and VarDict v.1.5.1[31]. We used a variant allele frequency (VAF) cutoff of 2% in accordance with Steensma et al.[32], and we excluded variants with a minor allele frequency >1% in the ExAC (v. 0.3), TOPMED (Freeze 3), and 1000 Genomes (Phase 3v. 5c) databases, as well as rare variants with VAF between 40% and 60% which were concordant between twins.

**DNA methylation data and quality control (LSADT CHIP cohort)**. DNA methylation data were available for a subset ($N = 309$) of the individuals included in our previous sequencing study. Methylation levels in whole blood were measured using the Illumina Infinium Human Methylation 450k BeadChip array (450k array) according to the manufacturer's protocol. For quality control we used MethylAid v. 1.26.0[33] to search for outlier and low quality samples, but found none. We excluded probes that had (a) zero signal in one or more samples, (b) <3 beads, (c) detection $P$ value > 0.01, or (d) call rate <0.95 across either samples or probes. Furthermore, we excluded sex chromosome probes and cross-reactive probes according to Chen et al.[34], and probes affected by single-nucleotide polymorphisms using the function dropLociWithSnps from the R package minfi v. 1.34.0[35]. After probe-level QC we performed functional normalization[36] using four principal component analysis (PCA) components. At the final step of quality control, we conducted a new PCA and identified four samples which were outliers in the first seven components and thus removed from subsequent analyses. Immune cell deconvolution was performed according to Houseman et al.[37] using the minfi function EstimateCellCounts.

**DNA methylation data and quality control (CCUS patients and healthy controls)**. Peripheral blood granulocyte DNA samples from five CCUS patients and eight healthy controls were isolated using standard Ficoll Plaque PLUS (GE healthcare) gradient separation. Granulocyte DNA extraction was performed using AllPrep DNA/RNA/miRNA Universal kits (Qiagen). Bone-marrow mononuclear cells from 20 CCUS patients were isolated using standard Ficoll Plaque PLUS (GE healthcare) gradient separation and subsequently T cell-depleted using RoboSep Human CD3 Positive Selection Kit II (StemCell Technologies, Vancouver, Canada) and RoboSep #20000 (Software version 4.6.0.1) (StemCell Technologies) with a

customized protocol using two-quadrant separation. DNA was extracted using QIAamp DNA Blood mini Kit (Qiagen, Hilden, Germany). All CCUS samples were analyzed using the Illumina HumanMethylationEPIC BeadChip.

For both CCUS datasets, probes with detection $P$ values > 0.01, bead count <3 in at least 5% of samples, non-CpG sites, probes targeting sex chromosomes, SNPs <5 bp from the target CpG[38] and probes which previously showed binding to multiple target CpGs[39,40] were excluded using the ChAMP R package v. 2.18.3[41], leaving 736,088 and 721,527 remaining CpG sites (granulocyte and MNC data respectively). Data was normalized for type 1 and type 2 probes using Swan normalization in the minfi package[42], and ComBat (R package sva v. 3.36.0) was used for sample plate correction.

**Statistics**. Statistical analyses were done using R versions 3.5.2 and 4.0.2 unless otherwise stated.

Methylation beta values were logit transformed to $M$ values unless otherwise stated. In the CHIP cohort, association between methylation levels at each CpG site and mutation status (*TET2* mutation vs no CHIP mutations and *DNMT3A* mutations vs no CHIP mutations) was analyzed using the lmerTest v. 3.1-3 R package in a multivariate linear mixed effects regression with methylation $M$ value as the outcome and mutation status as the explanatory variable, with adjustment for age, sex, methylation profiling batch, Houseman cell types (granulocytes, monocytes, CD4$^+$ T cells, B cells), and the first four principal components. To account for similarity between twins we included twin pair as a random intercept. In the CCUS and AML datasets, differential methylation by *TET2* mutation was analyzed using linear models using the Limma package (v. 3.44.3)[43] with adjustment for age and sex. The genomic inflation factor, $\lambda$, was calculated as the observed median $\chi^2$ statistic divided by the expected median $\chi^2$ statistic. For each epigenome-wide analysis in CHIP, CCUS, and AML, we selected the threshold for statistical significance as the $P$ value below which 95% of sites were hypermethylated (hypomethylated for *DNMT3A*). These were $1.4 \times 10^{-5}$ for *TET2* mutations in CHIP, $4.1 \times 10^{-5}$ for *DNMT3A* mutations in CHIP, and $7.7 \times 10^{-6}$ for *TET2* mutations in CCUS granulocyte data. CpG sites below this threshold and with a positive association with *TET2* mutations were defined as significantly hypermethylated. In AML *TET2* analyses, there was no threshold below which 95% of sites were hypermethylated, so we instead selected the 2000 most significantly hypermethylated sites for further investigation. Associations between dichotomous outcomes were tested using $\chi^2$ test, unless the expected value of one or more cells in the contingency table was below five, in which case Fisher's exact test was used. All reported $P$ values are two-sided.

Prediction of *TET2* mutation status was done using leave-one-out cross validation of an elastic net regression model with the R package glmnet v. 4.1. At each iteration the optimal lambda parameter was first calculated using the function cv.glmnet and then a regression model was fitted using the selected lambda.

No patient samples were analyzed more than once.

**Gene ontology, chromatin, and transcription factor motif analysis**. Gene Ontology term enrichment analysis was done using GREAT[44] by treating each selected CpG site as a region of length 1 bp. Enriched Gene Ontology terms were filtered based on statistical significance (Bonferroni-corrected $P < 0.05$) in both the region-based binomial test and the gene-based hypergeometric test. CpG site relation to CpG islands was annotated using the Illumina documentation file for the 450k array. CpG site relation within enhancer elements was annotated using Bedtools v2.27.1 to the monocyte dataset of chromatin states from the Roadmap Epigenomics project (ChromHMM 15-state model)[18]. Motif enrichment analysis was performed with Analysis of Motif Enrichment (AME)[45], using the Fisher's exact test method. AME takes a set of ranked sequences and uses partition maximization to identify motifs that are enriched among the highly ranked sequences. The HOCOMOCO Human (v11 full)[46] database of transcription factor position weight matrices was used as input motifs, and input sequences were defined as 200 bp regions centered around each of the CpG sites from the 450k array which were located in known enhancer regions. Significant enrichment was defined as an enrichment ($E$) value <0.05. For the CHIP data, we had two separate estimates of the *TET2*-dependent change in methylation level at each CpG site, i.e. the effect estimates from each of the two regression analyses (*TET2* mutated vs no CHIP and *TET2* mutant clone size), and these two were combined into a composite score for each CpG site calculated as the average of the normalized effect estimates, all 200 bp regions were then ranked according to this score in the input to AME.

**Reporting summary**. Further information on research design is available in the Nature Research Reporting Summary linked to this article.

## Data availability
The raw and processed DNA sequencing and methylation data in the CHIP cohort generated in this study as well as other individual-level data from the Danish Twin Registry (DTR), including those derived from the Danish National Patient Registry, are are only available under restricted access, since these data are considered sensitive personal data according to Danish Law and the European Union General Data Protection Regulation (GDPR) and thus cannot be shared with third-parties without prior approval. To access the CHIP dataset, an application must be sent to tvilling@health.sdu.dk. The

application will then be processed by the Scientific Board at DTR, and subsequently by Legal Services at the University of Southern Denmark (SDU-RIO) or the Danish Data Protection Agency (DDPA). Access can only be granted for research purposes, and only if a data processor or data transfer agreement can be made in accordance with Danish and European law at the given time. The expected timeframe from request to decision is ~6 weeks at the DTR and a few months at the SDU-RIO and DDPA.

The raw and processed DNA sequencing and methylation data on CCUS patients generated in this study are available under restricted access for the same reasons as stated for the CHIP dataset. To access the CCUS dataset, an application must be sent to Kirsten.Groenbaek@regionh.dk. Access can only be granted for research purposes, and only if a data processor or data transfer agreement can be made in accordance with Danish and European law at the given time. The expected timeframe from response until access is granted is ~6 months.

The chromatin state dataset for monocytes and HCSs used in this study are available in the Roadmap Epigenomics database under epigenome ids E029 and E035 (core 15-state model) [https://egg2.wustl.edu/roadmap/web_portal/chr_state_learning.html]. The transcription factor binding models dataset used in this study is available in the HOCOMOCO database (full dataset) [https://hocomoco11.autosome.ru/human/mono?full=false]. The TCGA LAML DNA sequencing and methylation datasets used in this study are available from the Genomic Data Commons Data Portal under project ID TCGA-LAML [https://portal.gdc.cancer.gov/]. The ExAC dataset (version 0.3) used in this study is available from the UCSC Genome Browser (https://hgdownload.soe.ucsc.edu/gbdb/hg19/ExAC/ExAC.r0.3.sites.vep.hg19.vcf.gz). The 1000 Genomes dataset (Phase 3 v5c) used in this study is available from the 1000 Genomes web page (http://ftp.1000genomes.ebi.ac.uk/vol1/ftp/release/20130502/ALL.wgs.phase3_shapeit2_mvncall_integrated_v5c.20130502.sites.vcf.gz). The TOPMED dataset (Freeze 3) used in this study is available from the BRAVO variant browser (https://bravo.sph.umich.edu/freeze3a/hg19/download).

Source Data are provided with this paper. All results from analyses of differential methylation by DNMT3A or TET2 mutations in CHIP and CCUS (regression coefficients, P values, mean values in cases and controls) and the complete results from the TF motif enrichment analyses are provided as Supplementary Data with this paper. Source data are provided with this paper.

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

## Acknowledgements

M.T. and L.G. are supported by the Research Foundation at Rigshospitalet. The Danish Twin Registry is supported by the Odense University Hospital AgeCare program (Academy of Geriatric Cancer Research). The Danish Twin Registry has been supported

by The National Program for Research Infrastructure 2007 grant 09-063256 from the Danish Agency for Science Technology and Innovation, the Velux Foundation, and the National Institutes of Health, National Institute on Aging grant P01 AG08761. K.G., J.W., and K.H. are supported by the Novo Nordisk Foundation (Novo Nordisk Foundation Center for Stem Cell Biology, DanStem; grant NNF17CC0027852). J.W. is also supported by the Novo Nordisk Foundation grant NNF200C0060141. K.G. is supported by The Danish Cancer Society (Danish Research Center for Precision Medicine in Blood Cancer; grant 223-A13071-18-S68) and from Greater Copenhagen Health Science Partners (Clinical Academic Group in Blood Cancers).

## Author contributions

M.T., M.S., J.W.H., J.W., K.C., and K.G. designed the study. K.C and K.G. provided study samples (Twin Registry and CCUS patients, respectively). J.W.H. handled DNA sequencing of twin and CCUS samples. M.T. and M.S. analyzed CHIP data. L.G. and K.K. gathered and handled CCUS blood and bone marrow samples and cell separation. L.G., M.N., and M.T. analyzed CCUS data. M.T. analyzed TCGA data. M.S., J.W., K.H., and K.G. supervised bioinformatics and statistical analyses. M.T. wrote the first draft of the paper. All authors edited the paper and approved the final version.

## Competing interests

The authors declare no competing interests.
