## [Peer Review File · Nature Communications]

TET2 mutations are associated with hypermethylation at key regulatory enhancers in normal and malignant hematopoiesisREVIEWER COMMENTS

Reviewer #1 (Remarks to the Author): Expert in cancer epigenomics and bioinformatics

Tulstrup et al. describe the methylation landscape in clonal hematopoiesis of indeterminate potential (CHIP) and clonal cytopenia of undetermined significance (CCUS) with TET2 mutations. They observed a strong hypermethylation pattern which largely overlaps with those in AML patients. The hypermethylation affects mainly enhancers and overlaps with the binding site of important transcription factors.

The manuscript is well written and easy to follow. The data analysis is comprehensive and precisely conducted. The tools are adequate. The findings of the paper are interesting, however, I am not fully convinced about its translational value, as the paper stays descriptive. On the other hand, the authors describe the possible implications of their findings well in the Discussion.

The authors suggest that the hypermethylation patterns in CHIP might represent an early event towards AML. Do the authors see an association between clone size and the level of hypermethylation? This would further strengthen their hypothesis.

Did the authors consider looking into the cell-type composition of the samples? I see it described in the methods, but I didn't see the results. I am aware that the TET2-related hypermethylation might disrupt the analysis, but it would be interesting to see if there is any shift in the composition.

Minor comments:

The Supplementary Figures 8 and 9 don't fully reflect on their description in the main text.

Reviewer #2 (Remarks to the Author): Expert in genomics and epigenomics of myeloid malignancies

This manuscript reports on the genome-wide analyses DNA methylation in samples from clonal hematopoiesis and malignant samples carrying or not somatic mutation in TET2 and DNMT3A genes. It is carefully performed and essentially confirms that the main consequence of TET2 inactivation is hyper methylation of "enhancer" regions. In TET2-mutant samples hypermethylated regions are associated with ETS-family and CEBP family specific DNA binding sequences.

What is the blood cell composition of the analyzed samples? Could some aspects of the differential methylation profile be due to enrichment in a given cellular fraction? Does the fact that ccus samples appear to be granulocytes bias the methylation profile data, with respect to other samples?

Cellular transformation processes are usually gradual. It may be worth to describe more thoroughly the differences in methylation profile between CHIP, Ccus and AML, since the thresholds used may mask some hints.

Figure 5 shows essentially differentially methylated CpGs. How many enhancer regions do they represent? To what extent are the HSC-enhancers partially methylated in ccus?

In other words, do the presented data allow describing a hypermethylation path leading from CHIP to ccus and AML?

The fact that only a minority of TSS is hypermethylated does not formally exclude that it participates to cellular transformation. Indeed, there are examples of promoter methylation in cancer (e.g. PMID: 29275866, PMID: 26876596). If identified, the consequences of promoter methylation could be validated by analyzing transcription data. In addition Izzo et al (ref 9, PMID: 32203468) analyzed marrow progenitor cells, and not blood cells, so the context is different.

Would a deconvolution approach shed light on some of these aspects (cellular composition and transformation processes)?

Is there any overlap between mutant-DNMT3A and mutant TET2 differentially methylated CpG/profiles? Why were mutant DNMT3A AML samples not included in the analyses and compared to CHIP samples?

What are candidate transcription factors sequences at hyper methylated HSC enhancers observed in AML? (Fig 5)

Reviewer #3 (Remarks to the Author): Expert in clonal haematopoiesis and leukaemia genomics

In the current study Tulstrup et.al used the 450k methylation array to study methylation patterns across ~300 individuals either healthy with no CHIP, or healthy individuals with CHIP and also few CCUS cases and re-analysis of the TCGA AML data. Their main conclusion was that TET2 mutations among individuals with CHIP and CCUS are associated with DNA hypermethylation in a non-random, and was identified specifically in enhancer elements, non-CpG island regions of genes involved in myeloid leukocyte function and immune response. These results were in line with recent single cell RNA-seq and single cell methylation analysis of mice with TET2 knockout mice (Izzo F et.al Nat gene) and few other studies.

The authors' main claim was that TET2 methylation consequences were not extensively studied among healthy individuals.

Major comments:

1. TET2 haploinsufficiency has been studied among healthy individuals who carries a frameshift deletion as a germline mutation and the methylation patterns of these individuals has been extensively studied (<https://doi.org/10.1038/s41467-019-09198-7> | www.nature.com/naturecommunications).

In this study they performed whole genome BS sequencing on a smaller cohort however well controlled as the WT individuals came from the same pedigree and were aged matched. Based on the hypermethylated regions identified CHIPseq analysis has been performed to validate the hypothetical TFs binding to these regions and better describe chromatic modifications. Altogether, in this study similar conclusions have been drawn.

2. Technical issue the measurement of whole blood methylation what are we actually measuring??? The fact that the authors could not identify any DNA methylation differences in the DNMT3a cases might suggest that using the mature peripheral blood might not be the right way to study the consequences of TET2 and DNMT3a on methylation, and more defined lineages should be studied based on the question asked. CHIP is a complex phenomenon with possible defects in many lineages. The methylation changes in stem cell cause increased self renewal while the effects in other lineages are less clear and might be different in different cell types as was demonstrated in Izzo et.al Nat Gen. Accordingly looking on the bulk whole blood might be misleading and not very informative.

3. Measuring DNA methylation from whole blood has potential biases:

It has been shown that whole blood DNA methylation basically correlates with blood counts one example is here <http://dx.doi.org/10.4161/epi.25430> but there are more. These means that the hypermethylated regions identified in the current study could potentially be the result of a specific lineage which is more abundant among TET2 individuals. Indeed studies have demonstrated that in CHIP one of the only mutations with changes in blood counts is TET2 (Buscarlet M, Blood 2017). Accordingly the fact that methylation changes were identified in TET2 and not DNMT3A could be due to changes in blood counts. Controlling for blood counts is important.

A second bias of whole blood methylation analysis is the effect of Smoking PMID: 26478754 Which was not controlled here.

General remarks:

All three reviewers very relevantly ask about the blood cell compositions of the CHIP samples. In the first version of the manuscript, we inferred cell-types using the most widely used blood cell deconvolution method (Houseman et al. BMC Bioinformatics 2012), as is standard in epigenomics analyses of whole blood. However, we only used these measures to adjust for any potential confounding from shifts in cell-type proportions on DNA methylation changes. We have now included a figure (Supplementary Figure 5) to show these estimated cell type proportions. *TET2* mutated individuals had slightly higher estimated monocyte levels, but otherwise we observed no statistically significant differences (*DNMT3A* mutated individuals had slightly lower estimated monocyte levels just above the significance threshold at $P = 0.05$). We have added a sentence about this in the main text (lines 78-83)

The adjustment for cell-type proportions in our analyses on the CHIP dataset allows us to conclude that the observed changes in methylation levels are very unlikely to be caused by changes in blood cell proportions. We have now further stressed this point in the manuscript (lines 87-88 and 110-112). However, based on the CHIP dataset alone we cannot conclude whether the observed changes are general or lineage-specific. That is to say, for any given CpG site that is hypermethylated in *TET2*-mutated individuals, we cannot from whole-blood analyses infer whether the hypermethylation occurs in a specific cell type or in many/all cell types. Our analyses in CCUS patients in the first manuscript were restricted to peripheral blood granulocytes, so at least we could conclude that the hypermethylation signature observed in whole blood in CHIP is also clearly detectable in granulocytes in CCUS. To further address the question whether the observed changes are cell type-specific, we have now included data from 20 new CCUS patients, of whom 10 had a *TET2* mutation. Here, we performed methylation array analysis of T cell-depleted bone marrow-derived mononuclear cells (MNCs), which reproduced the same hypermethylation signature in this subset of cells (see lines 150-158 and new Supplementary Figure 11).

In the future, it will be exciting to see more detailed analyses of lineage-specific effects of *TET2* mutations. We have now stressed this point in the discussion (lines 323-327)

We have added one new co-author, Katja Kaastrup, who gathered and handled the CCUS MNC samples including cell sorting, T-cell depletion, and preparation for methylation analysis.

Point-by-point responses (reviewer comments in blue):

Reviewer #1 (Remarks to the Author): Expert in cancer epigenomics and bioinformatics

Tulstrup et al. describe the methylation landscape in clonal hematopoiesis of indeterminate potential (CHIP) and clonal cytopenia of undetermined significance (CCUS) with *TET2* mutations. They observed a strong hypermethylation pattern which largely overlaps with those in AML patients. The

hypermethylation affects mainly enhancers and overlaps with the binding site of important transcription factors.

The manuscript is well written and easy to follow. The data analysis is comprehensive and precisely conducted. The tools are adequate. The findings of the paper are interesting, however, I am not fully convinced about its translational value, as the paper stays descriptive. On the other hand, the authors describe the possible implications of their findings well in the Discussion.

It is true that an important goal of our paper is to provide new insights into the role of TET2 and somatic *TET2* mutations in normal and malignant hematopoiesis. While many of these observations have no direct translational value at the moment, some of our results do have direct and important consequences for other fields of epigenetics research, where *TET2* mutations could be an important confounder between age-related phenotypes and blood DNA methylation. We think it is important that researchers in other branches of epigenetics research become aware of this phenomenon.

Another point which we did not mention in the first version of the manuscript is that the methylation levels at the *TET2* mutation-associated CpG sites may in future studies serve as a marker for TET2 activity. We have added a discussion of this in lines 340-348.

The authors suggest that the hypermethylation patterns in CHIP might represent an early event towards AML. Do the authors see an association between clone size and the level of hypermethylation? This would further strengthen their hypothesis.

We would like to thank the reviewer for this very relevant question. We tested this, and we did indeed find a positive association between clone size (VAF) and methylation level for 98.3% of the 2,741 significantly hypermethylated sites in CHIP. We have included this in the main text (lines 97-99) and shown the associations in Supplementary Figure 6.

In the two small sets of CCUS patients, the VAFs were narrowly distributed around 40% and 50%, respectively, and thus did not allow for meaningful modeling of associations with clone size.

Did the authors consider looking into the cell-type composition of the samples? I see it described in the methods, but I didn't see the results. I am aware that the TET2-related hypermethylation might disrupt the analysis, but it would be interesting to see if there is any shift in the composition.

This is a very good and relevant point. Please see general remarks above.

Minor comments:

The Supplementary Figures 8 and 9 don't fully reflect on their description in the main text.

We have added two supplementary tables to clarify how the models classified the *TET2* mutations. (Supplementary Tables 7 and 8).

Reviewer #2 (Remarks to the Author): Expert in genomics and epigenomics of myeloid malignancies

This manuscript reports on the genome-wide analyses DNA methylation in samples from clonal hematopoiesis and malignant samples carrying or not somatic mutation in *TET2* and *DNMT3A* genes. It is carefully performed and essentially confirms that the main consequence of *TET2* inactivation is hypermethylation of "enhancer" regions. In *TET2*-mutant samples hypermethylated regions are associated with ETS-family and CEBP family specific DNA binding sequences.

What is the blood cell composition of the analyzed samples? Could some aspects of the differential methylation profile be due to enrichment in a given cellular fraction? Does the fact that ccus samples appear to be granulocytes bias the methylation profile data, with respect to other samples?

Please see the general remarks above. Since we adjusted for cell composition in the CHIP analyses, the observed differences are unlikely to be caused by enrichment of a given cellular fraction. In our opinion, the granulocyte-specific analyses do not introduce bias, but rather they add more detail than whole-blood measurements, especially when interpreted in combination with our new analyses of MNCs in CCUS.

Cellular transformation processes are usually gradual. It may be worth to describe more thoroughly the differences in methylation profile between CHIP, Ccus and AML, since the thresholds used may mask some hints.

Please see below

Figure 5 shows essentially differentially methylated CpGs. How many enhancer regions do they represent?

The two categories HSC-specific enhancers and monocyte-specific enhancers represent 14,398 and 14,017 enhancer regions, respectively. We have added this to the figure legend.

To what extend are the HSC-enhancers partially methylated in ccus?

This is an interesting question, which we assume refers partly to the findings presented in Figure 5E. The figure shows that at HSC-specific enhancers, the mean difference in methylation between *TET2* mutated cases and *TET2* WT controls is 0.19 on an M value scale (logit-transformed beta values). This only shows that *TET2* mutations increase the methylation levels at these sites slightly, but it does not necessarily relate to the absolute methylation level at HSC-specific enhancers in *TET2* mutated or WT samples. In fact, the CpG sites in these enhancers display a wide range of methylation levels in the control cohort. Calculated on a beta value scale, the mean increase in methylation in *TET2*-mutated cases is only 0.022 (but with a wide distribution and some sites showing beta value changes upwards of 0.4). The main conclusion from these results is that in CCUS *TET2* affects methylation levels at different (HSC-specific enhancer) regions, indicating that the site-specific methylation effect of *TET2* is altered in CCUS compared to CHIP and that *TET2* mutations have effects on different parts of the epigenome. Secondly, it can be speculated that these methylation changes over time contribute to altered gene expression and disease progression, but as this reviewer correctly points out, cellular transformation processes are slow and gradual, and in the case of CCUS it may take many years before the disease progresses to MDS or AML (if ever). It would be very exciting to follow the methylation pattern longitudinally in CCUS patients and see whether e.g. increasing methylation at HSC-specific enhancers is a predictor of disease progression, but we consider that to be beyond the scope of this study.

In other words, do the presented data allow describing a hypermethylation path leading from CHIP to CCUS and AML?

We show that during the development from CHIP to CCUS and AML, *TET2*-associated hypermethylation shifts from one set of enhancers to another (Figure 5E).

Direct comparisons between the methylation levels in the different datasets are not possible due to the different batch effects; derived from different sample types, different sample preparations, different array platform and timepoints. To address the question without these confounders, we have now tried to analyze the development in hypermethylation in another way: We selected all CpG sites (N = 328,792) that were present in all three datasets (CHIP, CCUS blood granulocytes, AML) and calculated the mean methylation level (beta value) in *TET2* mutated cases in each dataset. To avoid batch effects, we categorized the beta values into tertiles for each dataset (low, medium, or high, see Additional File for Review, Figure 1A). We then defined methylation trajectories for all CpG sites, i.e. we grouped sites according to their methylation category in each of the three datasets CHIP -> CCUS -> AML. For example, a site that is in the “medium” category in both CHIP and CCUS, but in the “high” category in AML follows the trajectory “medium -> medium -> high”. We then fitted a logistic regression model with the hypermethylated sites in *TET2* mutated CHIP (2,106 of 2,741 sites present) as the outcome and all the different methylation trajectories as the explanatory variable (Additional file for review, Figure 1B). The two most enriched CpG trajectories in the 2,106 hypermethylated sites were “medium -> medium -> high” (391 sites) and “medium -> high -> high” (167 sites) (ORs 3.2 and 2.6, respectively, both $P < 1 \times 10^{-16}$). In summary, we find that the most dominant path is for methylation sites that are at medium or high level of CpG methylation in all three data set. While potentially interesting, we find that this observation does not add clear novelty to our manuscript but warrant further and more detailed investigation of longitudinal dataset from patients progressing from CHIP to AML.

The fact that only a minority of TSS is hypermethylated does not formally exclude that it participates to cellular transformation. Indeed, there are examples of promoter methylation in cancer (e.g. PMID: 29275866, PMID: 26876596). If identified, the consequences of promoter methylation could be validated by analyzing transcription data. In addition Izzo et al (ref 9, PMID: 32203468) analyzed marrow progenitor cells, and not blood cells, so the context is different.

Would a deconvolution approach shade light on some of these aspects (cellular composition and transformation processes)?

It is true that our results do not formally exclude that TSS hypermethylation can influence the cellular phenotype in *TET2* mutations. Unfortunately, we do not have transcription data for these samples. We instead chose to search for hypermethylated TSS regions with *comb-p*, a commonly used algorithm for identifying differentially methylated regions. We investigated whether there were any overlapping regions between the CHIP, CCUS, and AML datasets, but only identified one small region that overlapped between CHIP and CCUS. *Comb-p* did not identify any significantly hypermethylated regions in AML. We describe these analyses in the Results (lines 159-162), Supplementary Methods 2, and Supplementary Table 6.

Importantly, the referenced paper by Kunimoto et al (PMID 29275866) specifically showed that *TET2* mutations alone did not cause methylation changes in the mouse *Spry2* promoter, but hypermethylation only occurred when *TET2* and *NRAS* were co-mutated. Similarly, Scourzic et al (PMID 26876596) only investigated cells in which *TET2* and *DNMT3A* were mutated together. Thus, to our knowledge there is

no published evidence that *TET2* mutations alone cause promoter hypermethylation. It would have been interesting to investigate co-occurrences of *TET2* and *DNMT3A* mutations, but there were not enough overlapping cases in our data (4 in CHIP, none in AML)

Why were mutant *DNMT3A* AML samples not included in the analyses and compared to CHIP samples?

We thank the reviewer for pointing this out. We focused on *TET2* mutations, as they were the strongest phenotype on CpG methylation changes. However, we agree that an analysis of *DNMT3A* mutations and the impact on hypomethylation in AML adds additional novelty and insights. We have now performed a parallel analysis of *DNMT3A* mutated cases, which is presented in the main text lines 232-236 and Supplementary Figure 13.

Is there any overlap between mutant-*DNMT3A* and mutant *TET2* differentially methylated CpG/profiles?

We investigated this and found that while there is no overlap in CHIP, there is a considerable overlap in AML. See lines 238-247.

What are candidate transcription factors sequences at hyper methylated HSC enhancers observed in AML? (Fig 5)

To answer this very relevant question, we analyzed TFBS at the hypermethylated HSC enhancers in the *TET2*-mutated AML cases. We found a large number of TF motifs (238) to be enriched at hypermethylated at HSC enhancers in AML. In agreement with our findings in CHIP and CCUS, we found ETS factors to be enriched. We have added two sentences about this (lines 220-223), and added the full results to Supplementary Table 7.

Reviewer #3 (Remarks to the Author): Expert in clonal haematopoiesis and leukaemia genomics

In the current study Tulstrup et.al used the 450k methylation array to study methylation patterns across ~300 individuals either healthy with no CHIP, or healthy individuals with CHIP and also few CCUS cases and re-analysis of the TCGA AML data. Their main conclusion was that *TET2* mutations among individuals with CHIP and CCUS are associated with DNA hypermethylation in a non-random, and was identified specifically in enhancer elements, non-CpG island regions of genes involved in myeloid leukocyte function and immune response. These results were in line with recent single cell RNA-seq and single cell methylation analysis of mice with *TET2* knockout mice (Izzo F et.al Nat gene(and few other studies.

The authors' main claim was that *TET2* methylation consequences were not extensively studied among healthy individuals.

We have changed the wording in our manuscript to clarify that we focus on somatic mutations (lines 66 and 267).

Major comments:

1. *TET2* haploinsufficiency has been studied among healthy individuals who carry a frameshift deletion as a germline mutation and the methylation patterns of these individuals have been extensively studied (<https://doi.org/10.1038/s41467-019-09198-7> | www.nature.com/naturecommunications).

In this study they performed whole genome BS sequencing on a smaller cohort however well controlled as the WT individuals came from the same pedigree and were age matched. Based on the hypermethylated regions identified ChIP-seq analysis has been performed to validate the hypothetical TFs binding to these regions and better describe chromatin modifications. Altogether, in this study similar conclusions have been drawn.

We agree that this paper is important to mention when discussing our results, which we have now included in the Discussion (lines 281-284). In our opinion, our study complements and extends the findings by Kaasinen et al, which analyzes data from a few *TET2* germline mutation carriers, most of whom are related. Importantly, these mutations are germline and thus not only present in the hematopoietic compartment. As opposed to the very common somatic *TET2* mutations seen in CHIP, CCUS, and myeloid cancers, germline *TET2* mutations are exceedingly rare. Furthermore, as Kaasinen et al point out: “It is important to note that the exposure of the germline mutation carriers to effects of *TET2* loss is life-long and extreme, as compared to the typical CH setting of a minor somatic heterozygous *TET2*-mutant subclone—2% or higher variant allele frequency—emerging late in life” (From Kaasinen et al, Nature Communications 2019). Finally, our paper addresses a different question, namely what happens with the *TET2* hypermethylation signature during the clonal evolution from CHIP to CCUS and AML. This was not addressed in the paper by Kaasinen et al.

2. Technical issue the measurement of whole blood methylation what are we actually measuring???

The fact that the authors could not identify any DNA methylation differences in the DNMT3a cases might suggest that using the mature peripheral blood might not be the right way to study the consequences of *TET2* and DNMT3a on methylation, and more defined lineages should be studied based on the question asked. CHIP is a complex phenomenon with possible defects in many lineages. The methylation changes in stem cells cause increased self-renewal while the effects in other lineages are less clear and might be different in different cell types as was demonstrated in Izzo et al Nat Gen. Accordingly looking on the bulk whole blood might be misleading and not very informative.

This is an interesting point. Especially the fact that we did not observe more hypomethylated sites in DNMT3A-mutated CHIP may be due to the use of mature, unsorted blood. We discuss this topic in the revised manuscript (lines 323-327).

While we agree that studying lineage-specific effects of *TET2* mutations would provide further insights, we also believe that studying whole blood can be useful and informative. One important message of our paper is exactly that these methylation changes are in fact detectable in peripheral blood without any cell sorting, even in CHIP with small clone sizes. This is an important message in itself, since these changes may confound other analyses of methylation changes in whole blood, and they may be a marker of *TET2* activity, as we mention in the discussion. Furthermore, the observation that the same changes are detectable both in granulocytes and MNCs in CCUS indicates that this phenomenon is not restricted to one or a few lineages, but may rather be a general consequence of *TET2* mutations in blood cells. See also general remarks above.

3. Measuring DNA methylation from whole blood has potential biases:

It has been shown that whole blood DNA methylation basically correlates with blood counts one example is here <http://dx.doi.org/10.4161/epi.25430> but there are more. These means that the hypermethylated regions identified in the current study could potentially be the result of a specific lineage which is more abundant among *TET2* individuals. Indeed studies have demonstrated that in CHIP one of the only mutations with changes in blood counts is *TET2* (Buscarlet M, Blood 2017). Accordingly the fact that methylation changes were identified in *TET2* and not *DNMT3A* could be due to changes in blood counts. Controlling for blood counts is important.

We thank the reviewer for this comment. We did control for blood counts, but realize that it was not properly described, and we have now clarified this in the manuscript (lines 87-88 and 110-112). Please also see the general remarks on cell-type specific analyses above

A second bias of whole blood methylation analysis is the effect of Smoking PMID: 26478754

Which was not controlled here.

This is an important point. We have added sensitivity analyses for *TET2* and *DNMT3A* mutations in the CHIP dataset (see lines 99-104, 107-108, Supplementary Methods 1, and Supplementary Figures 7 and 10. We did not find any evidence of confounding from smoking. We have therefore chosen to keep the results that were not adjusted for smoking as the main results, since there are a few missing data points in the smoking variables (7 individuals without smoking history). Unfortunately, smoking history was not available in the CCUS patients. Theoretically, smoking could be a confounder in the analyses on granulocyte DNA, since the controls used here were healthy controls. In the CCUS MNC data, however, smoking is unlikely to be a confounder, since the controls also had CCUS, but just not *TET2* mutations. *TET2* mutations are not more strongly associated with smoking than other myeloid mutations (Dawoud et al, Leukemia 2020, PMID 32518416). Overall, the fact that the methylation signatures in CHIP were unaffected by adjustment for smoking and that the methylation signatures were present in both of the two CCUS datasets despite different control groups makes it highly unlikely that any meaningful proportion of our findings are caused by smoking.

REVIEWER COMMENTS

Reviewer #1 (Remarks to the Author):

The authors addressed all the questions I had before, therefore I find the manuscript to be acceptable for publication in its current form.

Reviewer #2 (Remarks to the Author):

The authors have satisfactorily addressed the reviewers' comments.

An additional question : is there any specificity in the " 2,000 TET2-associated 242 hypermethylated sites were enriched for hypomethylation in DNMT3A mutated tumors ($P < 1 \times 10^{-16}$, χ^2 243 test) " ?? (enhancers ? candidate transcription factors binding, function of the associated genes, etc..) ??

Reviewer #3 (Remarks to the Author):

Sorry but i am still not happy with the response.

DNA methylation of whole blood is highly biased and the fact that it is different between TET2 and controls is a fact however without understading whhter thisis due to a bias or real biology the current manuscript is problematic.

The best solution would be single cell methylation analysis with genotyping of mutant versus WT cells. An alternative would be methylation analysis of single cell colonies. The worse case is methylation analysis of sorted different cell populations for me it is hard to undestand the messge with out this.

At this point i would not accept the manuscript especially because it does not add much more from the Germline paper and the mice data.

Below are our responses to the reviewer comments: (our responses in blue)

Reviewer #1 (Remarks to the Author):

The authors addressed all the questions I had before, therefore I find the manuscript to be acceptable for publication in its current form.

We thank the reviewer for this assessment.

Reviewer #2 (Remarks to the Author):

The authors have satisfactorily addressed the reviewers' comments.

We thank the reviewer for this assessment.

An additional question : is there any specificity in the " 2,000 TET2-associated 242 hypermethylated sites were enriched for hypomethylation in DNMT3A mutated tumors ($P < 1 \times 10^{-16}$, χ^2 243 test) " ?? (enhancers ? candidate transcription factors binding, function of the associated genes, etc..) ??

We investigated this set of CpG sites, comprising a total of 1,639 sites. The results were highly similar and in agreement with our findings of all the TET2-associated hypermethylated sites with respect to gene ontology enrichment and to chromatin states (i.e. no enriched gene ontology terms and enrichment for enhancer regions). These observations are not surprising, as these CpG sites are simply a large subset (82%) of the 2,000 TET2-associated hypermethylated sites. With respect to TF motifs, our method for analyzing this does not allow us to investigate small subsets of genomic regions. An analysis of this question that would be comparable to the other motif enrichment analyses presented in the manuscript would require a methylation estimator for all enhancer CpG sites in the entire dataset (such as the mean scaled regression coefficient, which we used in the motif enrichment analyses of the CHIP data). While we do see the potential benefit of such an analysis, it is unfortunately not possible to obtain a biologically meaningful per-CpG score for sites associated with overlap between TET2 hypermethylation and DNMT3A hypomethylation with the data available.

Reviewer #3 (Remarks to the Author):

Sorry but i am still not happy with the response.

DNA methylation of whole blood is highly biased and the fact that it is different between TET2 and controls is a fact however without understanding whether this is due to a bias or real biology the current manuscript is problematic.

The best solution would be single cell methylation analysis with genotyping of mutant versus WT cells. An alternative would be methylation analysis of single cell colonies. The worse case is methylation analysis of sorted different cell populations for me it is hard to understand the message without this. At this point i would not accept the manuscript especially because it does not add much more from the Germline paper and the mice data.

We respectfully disagree with this reviewer's assessment of our results for several reasons.

First and foremost, our results have been adjusted for cell type composition using the Houseman algorithm (Houseman et al. BMC Bioinformatics 2012). This method is widely accepted within the epigenetics community as an appropriate method for eliminating confounding from shifts in cell type composition in blood-based methylation analyses. We also show that smoking does not confound our results.

Additionally, the observed methylation changes are present and highly similar in both granulocytes and mononuclear cells. Thus, even if there theoretically were some unmeasured confounding due to a specific, unmeasured cell type being more abundant in *TET2* mutated individuals (which we consider unlikely), our results still show that *TET2* mutations have the same effect on methylation levels in two mutually exclusive cell populations. Considering that blood granulocytes almost exclusively consist of neutrophils, we regard it as highly implausible that the changes in DNA methylation observed in granulocytes should be caused by a shift in cell type composition. Considering further that the same methylation changes were also observed in a completely different subset of cells, the suggestion of a bias from cell type composition is in our opinion not well-justified. From a biological perspective, we cannot imagine a realistic scenario in which a shift in cell type compositions causes highly similar methylation changes in two completely different and mutually exclusive cell populations.

The suggested single-cell methylation analyses would in principle be interesting to pursue, but such experiments are extremely costly and challenging, requiring joint DNA mutation and methylation analysis from patient-derived single cells. We believe this is both outside the scope of this work, and is not required nor necessary for the conclusions drawn in this paper.

Furthermore, in our opinion our paper adds important new insights into the effects of somatic *TET2* mutations in humans, which are not covered by the previous mouse and germline studies. We explain this in the Discussion and have also emphasized it in our previous response letter.

We have extended the section in the Discussion regarding the use of DNA methylation measurements from whole blood. We have added the following sentences:

"Our use of whole blood DNA methylation in the CHIP analyses has its limitations. Although the CHIP results are adjusted for estimated celltype compositions, we cannot exclude the possibility of unmeasured confounding from a shift in cell proportions not detected by the deconvolution algorithm. However, the results observed in CHIP were replicated in both CCUS granulocytes and MNCs, strongly suggesting that these methylation patterns are independent of cellular lineage."